# Fast Trajectory Tracking Control Algorithm for Autonomous Vehicles Based on the Alternating Direction Multiplier Method (ADMM) to the Receding Optimization of Model Predictive Control (MPC)

**DOI:** 10.3390/s23208391

**Published:** 2023-10-11

**Authors:** Ding Dong, Hongtao Ye, Wenguang Luo, Jiayan Wen, Dan Huang

**Affiliations:** 1School of Automation, Guangxi University of Science and Technology, Liuzhou 545036, China; 221068297@stdmail.gxust.edu.cn (D.D.); wgluo@gxust.edu.cn (W.L.); wenjyan@gxust.edu.cn (J.W.); 2Guangxi Key Laboratory of Automatic Detecting Technology and Instruments, Guilin University of Electronic Technology, Guilin 541004, China; 3Guangxi Key Laboratory of Automobile Components and Vehicle Technology, Guangxi University of Science and Technology, Liuzhou 545036, China; 4School of Mechanical and Automotive Engineering, South China University of Technology, Guangzhou 510641, China; dan78huang@scut.edu.cn

**Keywords:** trajectory tracking, model predictive control (MPC), alternately direction multiplier method (ADMM), quadratic programming, real-time performance

## Abstract

In order to improve the real-time performance of the trajectory tracking of autonomous vehicles, this paper applies the alternating direction multiplier method (ADMM) to the receding optimization of model predictive control (MPC), which improves the computational speed of the algorithm. Based on the vehicle dynamics model, the output equation of the autonomous vehicle trajectory tracking control system is constructed, and the auxiliary variable and the dual variable are introduced. The quadratic programming problem transformed from the MPC and the vehicle dynamics constraints are rewritten into the solution of the ADMM form, and a decreasing penalty factor is used during the solution process. The simulation verification is carried out through the joint simulation platform of Simulink and Carsim. The results show that, compared with the active set method (ASM) and the interior point method (IPM), the algorithm proposed in this paper can not only improve the accuracy of trajectory tracking, but also exhibits good real-time performance in different prediction time domains and control time domains. When the prediction time domain increases, the calculation time shows no significant difference. This verifies the effectiveness of the ADMM in improving the real-time performance of MPC.

## 1. Introduction

Autonomous driving is the main direction of development in the future automobile field. Its advantages have been demonstrated in many aspects and attracted wide attention and recognition from society. The main content of autonomous driving technology includes perception, decision-making, planning, control and other aspects [1]. Among them, trajectory tracking is the main content of the control level and one of the core technologies of autonomous vehicles, which determines the performance indicators such as the safety and comfort of vehicle operation [2].

Trajectory tracking is a fundamental function of autonomous vehicles that ensures that the vehicle travels along a preset path. Proportional integral derivative (PID) control, preview control theory, sliding mode control, deep learning, model predictive control (MPC), and other methods have provided numerous trajectory tracking strategies for autonomous vehicles [3,4,5,6,7,8]. MPC is widely used by researchers to address the trajectory tracking control problems for intelligent vehicles. MPC is characterized by predictive modeling, receding optimization, feedback correction, and the ability to handle constrained optimization problems [9,10,11,12,13]. Aiming at the trajectory tracking problem of distributed drive vehicles, based on the MPC and vehicle dynamics model, a method combining the trajectory tracking preview time control and differential torque control based on the reference angle has been proposed, which improved the trajectory tracking performance [14]. In order to improve the tracking ability of autonomous vehicles, fuzzy control is used in combination with MPC to improve the weight of the cost function in MPC through the fuzzy adaptive algorithm. This not only improves the accuracy of trajectory tracking, but also improves the steering smoothness of the vehicle [15]. Using the Kalman filter algorithm combined with the MPC controller, the online estimation of the vehicle’s nonlinear curvature response can reduce the wear and tear of vehicle components while ensuring comfort [16]. Feedforward compensation is integrated with MPC. The error from the expected path is calculated using a pure tracking algorithm as the input for feedback tracking control. The control output is then determined through MPC calculations, resulting in improved accuracy and robustness in trajectory tracking [17]. The weights of MPC are dynamically adjusted using a PSO-BP neural network, which improves the tracking performance of the autonomous vehicle under different speeds and road curvatures [18]. A vehicle model with steering dynamics is proposed. The cascaded MPC structure is used to separate the steering system of the vehicle dynamics model from the trajectory tracking controller. The simulation and test results show better performance than only considering the vehicle dynamics. However, integrating the steering system into the dynamics model can achieve optimal performance, but leads to higher computational requirements [19]. The above methods improve the tracking accuracy, but do not consider the complexity of MPC calculation.

MPC has significant advantages in solving trajectory tracking problems with various constraints. Extensive research has also been conducted to study the accuracy, stability, and robustness of vehicle trajectory tracking based on MPC [20,21]. However, MPC has a large online computation workload and low real-time performance. In addition, if the controlled model is too complex, it will significantly increase the MPC online iterative calculation, which will hinder the application of MPC in practice.

Therefore, improving the real-time performance of MPC has received increasing attention, and many research results have emerged [22,23]. A dynamic optimization toolkit can improve the computational speed and reduce the computation time of MPC [24,25]. A genetic algorithm is used to compute the optimal time domain parameters for real-time vehicle speed and road conditions, which improves the real-time performance of the controller [26]. But, it uses a relatively simple kinematic model. To improve the real-time performance of vehicle longitudinal speed planning, a nonlinear space-domain MPC (SMPC) is proposed to accelerate the nonlinear SMPC computation by generating thermal initialization and subsequently forming SMPC-RTI. However, it considers the longitudinal motion of the vehicle as well as the energy it saves [27]. In [28], the real-time performance and accuracy of vehicle trajectory tracking are improved by reducing the number of control steps or reducing the control frequency domain. Reducing the control frequency domain can meet the real-time requirements but may result in a slightly higher error compared to reducing the number of control steps. Using the linear parameter time-varying MPC and designing a linear quadratic regulator can reduce the computational cost of the dynamic control of the vehicle [29]. Alternating the direction multiplier method (ADMM) is favored by many researchers because of its good scalability. It has been successfully applied to machine learning, distributed computing, and other fields [30,31,32]. ADMM can divide the optimization variables into two parts and solve them in a separate framework, which can reduce the computational burden caused by the large scale of the system. In [33], a class of ADMM with nonlinear equality constraints is empirically studied and its convergence is analyzed. To enhance the real-time performance of the iterative linear quadratic regulator (iLQR), an optimization is conducted using the ADMM. When employing a logarithmic barrier function, this algorithm can circumvent the feasibility requirements of the initial trajectory during the first iteration, thereby expediting the optimization process [34]. It shows good real-time performance in different driving scenarios.

In summary, many methods have been proposed to improve the real-time performance of MPC, and some research results have been obtained. However, there are still some issues regarding improving the real-time performance of MPC for autonomous vehicle trajectory tracking. The combination of optimization algorithm and MPC can improve the real-time performance of MPC to a certain extent, but it will take up additional computational resources when solving the optimal parameters of MPC [34]. Simplifying the vehicle model can also improve the real-time performance of the controller, but it may not achieve ideal control effects in complex road conditions [28]. When using optimization algorithms or simplified models in conjunction with MPC controllers, the MPC problem is usually transformed into a quadratic programming (QP) problem [35]. The interior point method (IPM) or the active set method (ASM) are commonly used to solve the QP problem, but the ASM cannot utilize the sparsity of the MPC problem, and has to perform the active set operation at each iteration, which is suitable for the case of a small number of controls and constraints [36]. The IPM can utilize the sparsity of the MPC problem, but each iteration needs to solve the Karush–Kuhn–Tucker (KKT) system, and when the KKT system changes during the iteration, the solution process needs to decompose or inverse the matrix. When the number of constraints is large or the prediction time domain increases, the matrix dimension becomes larger and the computation time becomes longer [37]. This paper focuses on the problem of a long MPC solution time. Starting from the MPC problem-solving process, based on the framework of trajectory tracking control for autonomous vehicles, an auxiliary variable method is introduced to transform the quadratic problems in MPC into a separable structured optimization problem. The ADMM is used to solve it, and a decreasing penalty factor is used to ensure the convergence of the ADMM algorithm during the solution.

The organization of this article is as follows. The vehicle dynamic model and the MPC controller are presented in Section 2. The combination of the ADMM algorithm and MPC is introduced in Section 3. The simulation results of the controller are shown in Section 4. Finally, a brief conclusion is given in Section 5.

## 2. MPC-Based Trajectory Tracking Control

### 2.1. Vehicle Dynamics Model

To ensure that the vehicle follows its desired trajectory, the bicycle model in [38] is adopted to represent the vehicle dynamics. In this model, we do not consider the effects of suspension and aerodynamics, the state variables are defined as $\xi \left(t\right)={\left[\dot{x},\dot{y},\psi ,\dot{\psi },Y,X\right]}^{T}$, and the control input vector is the front wheel angle, defined as $u\left(t\right)=\delta_{f}$. The three-degrees-of-freedom dynamics model of the vehicle is shown in Figure 1. The *x* axis represents the longitudinal axis in the vehicle coordinate system, the *y* axis represents the lateral axis in the vehicle coordinate system, the xoy is the body coordinate system, and XOY is the ground coordinate system.

Differential equations for vehicle motion in lateral, longitudinal, and yawing directions are established based on Newton’s second law, as given below:(1)$$\left\{\begin{array}{c} m\ddot{x}=m\dot{y}\dot{\psi }+2F_{xf}+2F_{cr}\\ m\ddot{y}=-m\dot{x}\dot{\psi }+2F_{df}+2F_{lr}\\ I_{z}\ddot{\psi }=2aF_{df}-2bF_{lr} \end{array}\right.$$
where *m* is the mass of the vehicle, $I_{z}$ is the moment of inertia about the *z* axis, *a* and *b* are the distances between the center of mass and front/rear axles of the vehicle, ψ is the yaw angle of the vehicle, $F_{xf}$ and $F_{cf}$ represent the tire force on the longitudinal axis of the vehicle in the body coordinate system, and $F_{df}$ and $F_{lr}$ represent the tire force on the transverse axis of the vehicle in the body coordinate system, respectively.

When the lateral acceleration is not greater than 0.4 g, the longitudinal force and lateral force exerted on the tire can approximate a linear relationship as given below:(2)$$\left\{\begin{array}{c} F_{lf}=C_{lf}s_{f}\\ F_{lr}=C_{lr}s_{r}\\ F_{cf}=-C_{cf}\alpha_{f}\\ F_{cr}=-C_{cr}\alpha_{r} \end{array}\right.$$

In Equation (2), $C_{lf}$ and $C_{lr}$ represent the longitudinal stiffness of the front and rear tires of the vehicle, $C_{cf}$ and $C_{cr}$ represent the cornering stiffness of the front and rear tires of the vehicle, $s_{f}$ and $s_{r}$ represent the slip ratio of the front and rear tires of the vehicle, $\alpha_{f}$ and $\alpha_{r}$ represent the cornering angle of the front and rear tires of the vehicle, respectively. Assuming that both front wheel angles of the vehicle are equally small, the lateral acceleration satisfies the small angle assumption, and in this case, the following approximate relationship can be used:(3)$$\left\{\begin{array}{c} \alpha_{f}=\frac{\dot{y}+a\dot{\psi }}{\dot{x}}-\delta_{f}\\ \alpha_{r}=\frac{\dot{y}-b\dot{\psi }}{\dot{x}} \end{array}\right.$$

Finally, consider the transformed relationship between the vehicle body coordinate system and the earth inertial coordinate system as follows:(4)$$\left\{\begin{array}{c} \dot{X}=\dot{x}\sin\psi -\dot{y}\cos\psi \\ \dot{Y}=\dot{x}\sin\psi +\dot{y}\cos\psi \end{array}\right.$$

From Equations (1)–(4), we can obtain the following nonlinear dynamic model of intelligent vehicles:(5)$$\left\{\begin{array}{c} m\ddot{x}=m\dot{y}\dot{\psi }+2\left[C_{lf}s_{f}+C_{cf}\left(\delta_{f}-\frac{\dot{y}+a\dot{\psi }}{\dot{x}}\right)\delta_{f}+C_{lr}s_{r}\right]\\ m\ddot{y}=-m\dot{x}\dot{\psi }+2\left[C_{cf}\left(\delta_{f}-\frac{\dot{y}+a\dot{\psi }}{\dot{x}}\right)+C_{cr}\frac{b\dot{\psi }-\dot{y}}{\dot{x}}\right]\\ I_{z}\ddot{\psi }=2\left[aC_{cf}\left(\delta_{f}-\frac{\dot{y}+a\dot{\psi }}{\dot{x}}\right)-bC_{cr}\frac{b\dot{\psi }-\dot{y}}{\dot{x}}\right]\\ \dot{Y}=\dot{x}\sin\psi +\dot{y}\cos\psi \\ \dot{X}=\dot{x}\sin\psi -\dot{y}\cos\psi \end{array}\right.$$

It can be represented by the following differential equation:(6)$$\dot{\xi }=f\left(\xi \left(t\right),u\left(t\right)\right)$$

The state variables of the system are $\xi \left(t\right)={\left[\dot{x},\dot{y},\psi ,\dot{\psi },Y,X\right]}^{T}$, and the control variable is selected as $u\left(t\right)=\delta_{f}$.

### 2.2. Trajectory Tracking Based on Model Predictive Control

The nonlinear vehicle dynamics model developed in Equation (6) is complex and takes a long computational time to solve, so it is linearized using Taylor’s formula, omitting the higher-order terms except the first order. Assuming the Taylor expansion of Equation (6) at $\left[\xi_{0},u_{0}\right]$, the resulting linear time-varying equation is given as follows:(7)$$\dot{\tilde{\xi }}\left(t\right)=A\left(t\right)\tilde{\xi }\left(t\right)+B\left(t\right)\tilde{u}\left(t\right)$$
where $A\left(t\right)={\left.\frac{\partial f}{\partial \xi }\right|}_{\xi (i,u[i)}$, $B\left(t\right)={\left.\frac{\partial f}{\partial u}\right|}_{\xi (i,u[i)}$ are the Jacobi matrices.

Discretizing the above state space equations using the forward Euler’s method, the following equation can be obtained:(8)$$\left\{\begin{array}{c} \xi \left(k+1\right)=\left(TA\left(t\right)+I\right)\xi \left(k\right)+TB\left(t\right)u\left(k\right)+d_{k,t}\left(k\right)\\ d_{k,t}\left(k\right)=\xi_{0}\left(k+1\right)-\left(TA\left(t\right)+I\right)\xi_{0}\left(k\right)-TB\left(t\right)u_{0}\left(k\right) \end{array}\right.$$

From the above equations, the discrete linearized system can be shown:(9)$$\left\{\begin{array}{c} \xi \left(k+1\right)=A_{k,t}\xi \left(k\right)+B_{k,t}u\left(k\right)+d_{k,t}\left(k\right)\\ \eta (k)=C\xi (k) \end{array}\right.$$
where $A_{k,t}=I+TA\left(t\right)$, $B_{k,t}=TB\left(t\right)$, $C={\left[\begin{array}{cccccc} 0 & 0 & 1 & 0 & 0 & 0\\ 0 & 0 & 0 & 0 & 1 & 0 \end{array}\right]}^{\tau }$, $d_{k,t}\left(k\right)$ is the discrete state error of the system at *k*-time, and *T* is the sampling time.

Taking the increment of the control variable, i.e., Δu(k), as the input to the system reduces the effect of the sudden change in the control variable on the system, and therefore the original state vector needs to be augmented. Let the new state vector be $\bar{\xi }\left(k\right)={\left[\xi \left(k\right),u\left(k-1\right)\right]}^{T}$, we can obtain the new state space form as follows:(10)$$\left\{\begin{array}{c} \bar{\xi }\left(k+1|t\right)={\tilde{A}}_{k,t}\bar{\xi }\left(k|t\right)+{\tilde{B}}_{k,t}\Delta u\left(k|t\right)+\tilde{d}\left(k\right)\\ \bar{\eta }\left(k|t\right)=\tilde{C}\bar{\xi }\left(k|t\right) \end{array}\right.$$
where ${\tilde{A}}_{k,t}=\left[\begin{array}{cc} A_{k,t} & B_{k,t}\\ 0 & I \end{array}\right]$, ${\tilde{B}}_{k,t}=\left[\begin{array}{c} B_{k,t}\\ I \end{array}\right]$, ${\tilde{d}}_{k,t}=\left[\begin{array}{c} d_{k,t}\\ 0 \end{array}\right]$, $\tilde{C}=\left[\begin{array}{cc} C & 0 \end{array}\right]$.

The predicted output expression can be obtained as follows:(11)$$Y\left(t\right)=\Psi \left(t\right)\bar{\xi }\left(t\right)+\Phi \left(t\right)\Delta U\left(t\right)+E\left(t\right)D\left(t\right)$$
where $Y\left(t\right)=\left[\begin{array}{c} \xi \left(t\right)\\ \xi \left(t+1\right)\\ \vdots \\ \xi \left(t+N_{p}\right) \end{array}\right]$, $\Psi \left(t\right)=\left[\begin{array}{c} \tilde{C}{\tilde{A}}_{k,t}\\ \tilde{C}{\tilde{A}}_{k,t}^{2}\\ \vdots \\ \tilde{C}{\tilde{A}}_{k,t}^{N_{P}} \end{array}\right]$, $\Delta U\left(t\right)=\left[\begin{array}{c} u\left(t\right)\\ u\left(t+1\right)\\ \vdots \\ u\left(t+N_{p}\right) \end{array}\right]$, 

$D\left(t\right)=\left[\begin{array}{c} \tilde{d}\left(t\right)\\ \tilde{d}\left(t+1\right)\\ \vdots \\ \tilde{d}\left(t+N_{p}-1\right) \end{array}\right]$, $\Phi \left(t\right)=\left[\begin{array}{cccc} \tilde{C}{\tilde{B}}_{k,t} & 0 & \cdots & 0\\ \tilde{C}{\tilde{A}}_{k,t}{\tilde{B}}_{k,t} & \tilde{C}{\tilde{B}}_{k,t} & \vdots & 0\\ \vdots & \vdots & \ddots & \vdots \\ \tilde{C}{\tilde{A}}_{k,t}^{N_{p}-1}{\tilde{B}}_{k,t} & \tilde{C}{\tilde{A}}_{k,t}^{N_{p}-2}{\tilde{B}}_{k,t} & \cdots & \tilde{C}{\tilde{A}}_{k,t}^{N_{p}-N_{c}-1}{\tilde{B}}_{k,t} \end{array}\right]$, 

$E\left(t\right)=\left[\begin{array}{cccc} \tilde{C} & 0 & \cdots & 0\\ \tilde{C}{\tilde{A}}_{k,t} & \tilde{C} & \vdots & 0\\ \vdots & \vdots & \ddots & \vdots \\ \tilde{C}{\tilde{A}}_{k,t}^{N_{p}-1} & \tilde{C}{\tilde{A}}_{k,t}^{N_{p}-2} & \cdots & \tilde{C} \end{array}\right]$.

The primary objective of the MPC controller is to minimize the disparity between the output and the reference value. This ensures precise vehicle tracking along the desired path while maintaining lateral stability. However, due to the intricacies of the vehicle dynamics model and associated constraints, situations may arise where a numerical solution cannot be obtained within a single control cycle. To address this, a relaxation factor, denoted by ϵ, is introduced. This factor guarantees that a viable solution can be obtained in each control cycle. The objective function is expressed as follows:(12)$$\begin{array}{c} J\left(\bar{\xi }\left(t\right),u\left(t-1\right),\Delta U\left(t\right)\right)=\sum_{i=1}^{N_{p}}{∥\bar{\eta }\left(t+i|t\right)-\eta_{ref}\left(t+i|t\right)∥}_{Q}^{2}+\sum_{i=1}^{N_{c}-1}{∥\Delta U(t+i|t)∥}_{R}^{2}+\rho \epsilon^{2} \end{array}$$
where $N_{p}$ is the prediction time domain, $N_{c}$ is the control time domain, ϵ is the relaxation factor, ρ is the weight of the relaxation factor, $\bar{\eta }\left(t+i\right)$ represents the actual output of the system, $\eta_{ref}\left(t+i\right)$ represents the reference output, and ΔU(t+i) represents the increment of the forward turn angle.

According to Equation (12), the total cost of the objective function consists of three main parts. The first term relates to the cost associated with the deviation between the output trajectory and the reference trajectory. A greater deviation leads to a higher cost. The second term considers the cost associated with the forward corner increment. A larger forward corner increment during the tracking process results in a higher cost. Achieving a smoother control process is of the utmost importance when minimizing the control amplitude. The third term encompasses the relaxation factor and weight, ensuring the attainability of an executable solution for the objective function. When the vehicle is performing trajectory tracking, it is necessary to consider both the dynamic constraints of the vehicle itself and the limitations of the actuating mechanism. The constraints of vehicle dynamics are as follows:(13)$$\left\{\begin{array}{c} \Delta U_{\min}\leq \Delta U_{t}\leq \Delta U_{\max}\\ \Delta U_{\min}\leq A\Delta U_{t}+U_{t}\leq \Delta U_{\max}\\ y_{hc,\min}\leq y_{hc}\leq y_{hc,\max}\\ y_{sc,\min}-\varepsilon \leq y_{sc}\leq y_{sc,\max}+\varepsilon \\ \varepsilon >0 \end{array}\right.$$
where $y_{hc}$ is the hard constraint output and $y_{sc}$ is the soft constraint output.

Quadratic programming can solve the optimization problem of the MPC objective function; therefore, Equation (12) can be rewritten to the standard quadratic programming form, and the state variables and control variables constraint are introduced:(14)$$minJ=\frac{1}{2}{\left[\begin{array}{c} \Delta U(t)\\ \varepsilon \end{array}\right]}^{T}H_{t}\left[\begin{array}{c} \Delta U(t)\\ \varepsilon \end{array}\right]+G_{t}\left[\begin{array}{c} \Delta U(t)\\ \varepsilon \end{array}\right]+P_{t}$$
$$s.t.\left[\begin{array}{c} \Delta U_{\min}\\ \varepsilon \end{array}\right]\leq \left[\begin{array}{c} \Delta U\\ \varepsilon \end{array}\right]\leq \left[\begin{array}{c} \Delta U_{\max}\\ \varepsilon \end{array}\right]$$
where $H_{t}=2\left[\begin{array}{cc} \Phi_{t}^{T}Q\Phi_{t} & 0\\ 0 & \rho \end{array}\right]$, $G_{t}=\left[\begin{array}{cc} 2E{\left(t\right)}^{T}Q\Phi_{t} & 0 \end{array}\right]$, $Y_{ref}=\left[\begin{array}{ccc} \eta_{ref}\left(t\right) & \cdots & \eta_{ref}\left(t+N_{p}\right) \end{array}\right]$, $M=\left[\begin{array}{cccc} 1 & 0 & \cdots & 0\\ 1 & 1 & \cdots & 0\\ \vdots & \vdots & \ddots & \vdots \\ 1 & 1 & \cdots & 1 \end{array}\right]⊗I$, $E\left(t\right)=\Psi_{t}\xi \left(t\right)+E\left(t\right)D\left(t\right)-Y_{ref}\left(t\right)$, $P_{t}=E{\left(t\right)}^{T}QE\left(t\right)+U{\left(t-1\right)}^{T}RU\left(t-1\right)+\rho \varepsilon^{2}$.

Solving Equation (14) in each control cycle yields the optimal sequence of control increments:(15)$$\Delta U_{t}^{*}={\left[\begin{array}{cccc} \Delta u_{t}^{*} & \Delta u_{t+1}^{*} & \cdots & \Delta u_{t+N_{c}-1}^{*} \end{array}\right]}^{T}$$

If the first item in the control sequence is added to the system, the current control variable is given as follows:(16)$$u\left(t\right)=u\left(t-1\right)+\Delta u_{t}^{*}$$

After the whole system enters the next cycle, the system repeats the above process and updates the control sequence to complete the trajectory tracking of the intelligent vehicle. MPC obtains the optimal control sequence through iterative optimization search, and the actual application of the algorithm is limited when the iteration process is long. The ADMM algorithm has the factorization of dual ascent and the global convergence of a multiplier method, which has received widespread attention in recent years due to its low computational complexity and simple algorithm structure. In this paper, it is used to solve the optimization problem in the trajectory tracking MPC of autonomous vehicles.

## 3. Implementation of ADMM Algorithm for Trajectory Tracking MPC Problem

### 3.1. Alternating Direction Method of Multipliers

The alternating direction multiplier method is generally used to solve constraint programming problems with the following equation:(17)$$\left\{\begin{array}{c} \min\;f(z)+g(v)\\ s.t.\;Cz+Dv=b \end{array}\right.$$
where *f* and *g* are convex functions, *z* and *v* are the variables to be optimized, and $C\in R^{p\times n}$, $D\in R^{p\times m}$, $b\in R^{p}$, Cz+Dv=b are the linear equality constraints that the problem needs to satisfy.

By introducing the dual variable ω, the augmented Lagrangian function of the above equation is constructed as follows:(18)$$L_{\rho }\left(z,v,\omega \right)=f\left(z\right)+g\left(v\right)+\omega^{T}\left(Cz+Dv-b\right)+\frac{\rho }{2}{∥Cz+Dv-b∥}_{2}^{2}$$
where ρ>0 is the penalty parameter.

The iterative process of the ADMM algorithm includes three parts: iteratively updating the original variable *z*, iteratively updating the original variable *v*; and updating the process of the dual variable ω. The update strategy is given as follows [39]:(19)$$\left\{\begin{array}{c} z^{k+1}=\underset{z}{\arg\min}L_{\rho }\left(z,v^{k},\omega^{k}\right)\\ v^{k+1}=\underset{v}{\arg\min}L_{\rho }\left(z^{k+1},v,\omega^{k}\right)\\ \omega^{k+1}=\omega^{k}+\rho \left(Cz^{k+1}+Dv^{k+1}-b\right) \end{array}\right.$$

In general, it is convenient to use the scaling dual variable μ=ωωρρ to represent the iteration process:(20)$$\left\{\begin{array}{c} z^{k+1}=\underset{z}{\arg\min}\left\{f\left(z\right)+\frac{\rho }{2}{∥Cz+Dv^{k}-b+\mu^{k}∥}_{2}^{2}\right\}\\ v^{k+1}=\underset{v}{\arg\min}\left\{g\left(v\right)+\frac{\rho }{2}{∥Cz^{k+1}+Dv-b+\mu^{k}∥}_{2}^{2}\right\}\\ \mu^{k+1}=\mu^{k}+Cz^{k+1}+Dv^{k+1}-b \end{array}\right.$$

One advantage of the ADMM method is that it has only one parameter ρ, and under general conditions, the method can demonstrate convergence to all penalty factors. During the iterative process, the solution for the primal variables *z* and *v* are performed alternately, which reduces the scale of the problem and improves computational efficiency.

### 3.2. Model Predictive Controller Based on ADMM Improvement

Based on the previous section, this section provides a detailed explanation of the combination of the ADMM algorithm and trajectory tracking MPC problem. As shown in Figure 2, the ADMM algorithm is applied to the optimization and solving process in MPC. Based on the dynamic model of intelligent vehicles, the future outputs of the system are predicted using the state information of the vehicle and the control input, and then the feedback correction is performed using the actual output of the detection object. Finally, the ADMM algorithm is used to solve the optimization target online, and the current optimal control input is obtained.

To apply ADMM to the receding optimization process of MPC, the QP problem described by Equation (14) can be abbreviated as follows:(21)$$\left\{\begin{array}{c} \min J=\frac{1}{2}x^{T}Hx+f^{T}x\\ s.t.Ax\leq b \end{array}\right.$$
where $H=2\left[\begin{array}{cc} \Phi_{t}^{T}Q\Phi_{t} & 0\\ 0 & \rho \end{array}\right]$, $f={\left[\begin{array}{cc} 2E{\left(t\right)}^{T}Q\Phi_{t} & 0 \end{array}\right]}^{T}$, $x=\left[\begin{array}{c} \Delta U\\ \varepsilon \end{array}\right]$, $A=\left[\begin{array}{cc} I & 0\\ -I & 0\\ \Phi & 0\\ -\Phi & 0\\ M & 0\\ -M & 0 \end{array}\right]$, $b=\left[\begin{array}{c} \Delta U_{\max}\\ -\Delta U_{\min}\\ Y(t)-\Psi (t)\xi (t)-E(t)D(t)\\ Y(t)+\Psi (t)\xi (t)+E(t)D(t)\\ U{\left(t\right)}_{\max}-U\left(t\right)\\ U\left(t\right)-U{\left(t\right)}_{\min} \end{array}\right]$.

After formulating the quadratic programming in standard form for addressing the trajectory tracking challenge in autonomous vehicles, it is imperative to incorporate auxiliary variables and subsequently reformulate the problem into a structure conducive to the ADMM algorithmic resolution:(22)$$\left\{\begin{array}{c} \min_{x,z}J=f\left(x\right)+g\left(z\right)\\ s.t.Ax-b+z=0 \end{array}\right.$$
where $f\left(x\right)=\frac{1}{2}x^{T}Hx+f^{T}x$, g(z) is the indicator function. g(z) is defined as
$g\left(z\right)=\left\{\begin{array}{c} 0,\;if\;Az\leq b\\ +\infty ,\;otherwise \end{array}\right.$.

According to the multiplier method, the augmented Lagrangian function for the optimization problem can be obtained as follows:(23)$$L_{\rho }\left(x,z,y\right)=f\left(x\right)+g\left(z\right)+y^{T}\left(Ax-b+z\right)+\frac{\rho }{2}{∥Ax-b+z∥}_{2}^{2}$$

By introducing the dual scaling variable μ=yyρρ, according to Equation (14), the iterative update process of the ADMM algorithm can be obtained as follows:(24)$$\left\{\begin{array}{c} x^{k+1}=\underset{z}{\arg\min}\left\{\frac{1}{2}x^{T}Hx+f\left(x\right)+\frac{\rho }{2}{∥Ax-b+z^{k}+\mu^{k}∥}_{2}^{2}\right\}\\ z^{k+1}=\underset{v}{\arg\min}\left\{g\left(z\right)+\frac{\rho }{2}{∥Ax^{k+1}-b+z+\mu^{k}∥}_{2}^{2}\right\}\\ \mu^{k+1}=\frac{1}{\rho }y^{k+1}=\frac{1}{\rho }\left[y^{k}+\rho \left(Ax^{k+1}-b+z^{k+1}\right)\right]\\ =\mu^{k}+Ax^{k+1}-b+z^{k+1} \end{array}\right.$$

Compute the gradients of the original variable *x* and the auxiliary variable *z*, respectively. According to the first-order optimality condition, the iterative process formula is given as follows:(25)$$\left\{\begin{array}{c} x^{k+1}={\left(H+\rho A^{T}A\right)}^{-1}\left[-f-\rho A^{T}\left(z^{k}+\mu^{k}-b\right)\right]\\ z^{k+1}=\max\left\{0,-Ax^{k+1}-\mu^{k}+b\right\}\\ \mu^{k+1}=\mu^{k}+Ax^{k+1}-b+z^{k+1} \end{array}\right.$$

To accelerate convergence, a relaxation factor of α∈[1,2] is added, and the above iterative process can be obtained as follows:(26)$$\left\{\begin{array}{c} x^{k+1}=-{\left(H+\rho A^{T}A\right)}^{-1}\left[f+\rho A^{T}\left(z^{k}+\mu^{k}-b\right)\right]\\ z^{k+1}=\max\left\{0,-\alpha \left(Ax^{k+1}-c\right)+\left(1-\alpha \right)z^{k}-\mu^{k}\right\}\\ \mu^{k+1}=\mu^{k}+\alpha \left(Ax^{k+1}-b+z^{k+1}\right)+\left(1-\alpha \right)\left(z^{k+1}-z^{k}\right) \end{array}\right.$$

According to the first-order optimality condition of the QP problem, define the original residuals $s_{prim}$ and dual residuals $s_{dual}$ of (21), so we can obtain Equation (27):(27)$$\left\{\begin{array}{c} s_{prim}^{k+1}=Ax^{k}-b+z^{k}\\ s_{dual}^{k+1}=\rho A^{T}\left(z^{k+1}-z^{k}\right) \end{array}\right.$$

The algorithm convergence criterion is given according to the two residuals:(28)$$\left\{\begin{array}{c} {∥s_{prim}^{k}∥}_{2}\leq \varepsilon^{prim}\\ {∥s_{dual}^{k}∥}_{2}\leq \varepsilon^{dual} \end{array}\right.$$
where
(29)$$\varepsilon^{prim}=\sqrt{n}\varepsilon^{abs}+\varepsilon^{rel}\max\left\{{∥Ax^{k}∥}_{2},{∥z^{k}∥}_{2},{∥b∥}_{2}\right\}$$
(30)$$\varepsilon^{dual}=\sqrt{n}\varepsilon^{abs}+\varepsilon^{rel}\rho {∥A^{T}x^{k}∥}_{2}$$

In general, $\varepsilon^{rel}=10^{-3}$, $\varepsilon^{abs}$ can be selected according to the required accuracy.

The update of the ADMM algorithm variables requires the use of the results of the previous moment, and the optimal solution of the previous moment (d−1) is selected as the initial value of the algorithm at the current moment (d). The algorithm flow of the MPC solution control input Δu under trajectory tracking can be obtained.

(1)Initialize the MPC parameter to obtain the system status information at the *d*th moment.(2)According to the equation of the state variables of the system and the input and output variables, the objective function is converted into a quadratic programming problem in the form of Equation (14).(3)Rewrite Equation (14) to form as Equation (22) conforms to the ADMM solution.(4)The optimal solution obtained at time d−1 is used as the initial value of the solution to the time problem.(5)The variables are updated according to the iterative process of the ADMM algorithm, as shown in Equation (26).(6)According to Equations (27)–(30), to determine whether the iteration process meets the termination conditions, if it is met, stop the iteration, send the first term in the calculated optimal solution sequence to the control system as input, and enter step (7); if not, continue to iterate until the maximum number of iterations is reached.(7)Go to the next sampling moment d+1, and repeat step (1).

## 4. Simulation

To verify the effectiveness of the proposed algorithm, a joint simulation platform of Simulink and CarSim was built to simulate and validate the designed controller. The ASM and the IPM are commonly used methods for solving traditional MPC problems, and this paper compares and analyzes the proposed algorithm with ASM and IPM. The processor parameters of the laptop used in the simulation are AMD Ryzen 7 5700U with Radeon Graphics 1.80 GHz. The vehicle parameters used in simulation are shown in Table 1.

In the ordinary vehicle driving test, the double-shift condition is the test section with high frequency. Many scholars have also used it to test the trajectory tracking capabilities of autonomous vehicles. The desired trajectory is given by:(31)$$\left\{\begin{array}{c} Y_{ref}\left(X\right)=\frac{d_{y1}}{2}\left[1+\tanh\left(z_{1}\right)\right]-\frac{d_{y2}}{2}\left[1+\tanh\left(z_{2}\right)\right]\\ \varphi_{ref}\left(X\right)=\arctan\left[d_{y1}{\left(\frac{1}{\cosh\left(z_{1}\right)}\right)}^{2}\left(\frac{1.2}{d_{x1}}\right)-d_{y2}{\left(\frac{1}{\cosh\left(z_{2}\right)}\right)}^{2}\left(\frac{1.2}{d_{x2}}\right)\right] \end{array}\right.$$
where $z_{1}=\frac{2.4}{d_{x1}}\left(X-27.19\right)-1.2$, $z_{2}=\frac{2.4}{d_{x2}}\left(X-56.46\right)-1.2$, $d_{x1}=25$, $d_{x2}=21.95$, $d_{y1}=4.05$, $d_{y2}=5.7$.

The initial parameters of the algorithm are set as the road adhesion coefficient μ = 0.85, *v* = 20 m/s, relaxation factor α = 1.7, and the penalty factor ρ is a decreasing sequence with respect to time. The calculation time of the three algorithms is compared in the same and different prediction time domain and control time domain. The joint simulation diagram of the MPC trajectory tracking improved by ADMM is shown in Figure 3.

### 4.1. Comparison of Controllers under the Same Prediction and Control Horizon ($N_{p}\;=\;11$, $N_{c}\;=\;6$)

In this section, the traditional MPC is solved by ASM and IPM, and the simulation results of the proposed controller were analyzed under the same prediction and control time domains. The simulation results are shown in Figure 4, Figure 5 and Figure 6.

From Figure 4, it can be seen that, during trajectory tracking, both the MPC with ADMM algorithm improvement and the traditional MPC controller can achieve good tracking effect. The tracking accuracy of the MPC with the ADMM algorithm improvement is higher during the control time. We can also obtain an improved MPC that shows a better tracking accuracy, the proposed controller has a tracking error of 0.266 m at the maximum lateral displacement, while the other two controllers have a tracking error of 0.443 m. During the whole control time, we used the trajectory data driven by the vehicle and the expected trajectory data to obtain the root mean square error (RMSE) of each controller. The RMSE for improved MPC is 0.193, and it is 0.231 for both other controllers. The RMSE formula is shown below:(32)$$RMSE=\sqrt{\frac{1}{N}\sum_{i=1}^{N}{\left(Y\left(i\right)-Y_{ref}\left(i\right)\right)}^{2}}$$
where *N* represents the total simulation time divided by the sampling time, *Y* represents the actual trajectory of the vehicle, and $Y_{ref}$ represents the expected trajectory.

Figure 5 reflects the tracking of the three controllers on the yaw angle. It can be seen that all controllers show a good tracking effect on the change of yaw angle. At the longitudinal position of 80 m, the MPC with the ADMM algorithm improvement has some overshoot in tracking the desired yaw angle, and there are some oscillations at the longitudinal position of 90 m. It is worth noting that the driving comfort may be reduced due to the change in the heading angle, but it can quickly track the reference yaw angle afterwards.

Figure 6 reflects the calculation performance of three controllers. The initial iteration of ADMM takes a longer computational time, this is because, in the first iteration, due to the presence of constraints, the computation of the variables needs to solve a large system of linear equations, which will take a long time, and the next step is carried out alternatively to greatly reduce the computational time. The average computation time is shown in Table 2.

As shown in Table 2, the average computation time of the MPC improved by the ADMM algorithm is 0.0013 s, the average computation time of the ASM is 0.0020 s, and the average computation time of the IPM is 0.0035 s. Compared with the ASM, the average computation time of the MPC improved by the ADMM algorithm is reduced by 35%. The real-time performance of the MPC improved by the ADMM algorithm is higher than that of the IPM, with an average computation time reduction of 62.8%.

### 4.2. Comparison of Controllers under Different Control Horizons

In this section of the simulation, the control time domain of the MPC based on ADMM improvement is set to 10, and the control time domain of the traditional MPC is 4. The simulation results are shown in Figure 7, Figure 8 and Figure 9.

By choosing a prediction time domain of $N_{p}$ = 14, it can be seen from Figure 7 and Figure 8 that the trajectory tracking accuracy of the improved MPC is significantly better than that of the IPM and ASM. The error of the improved MPC at the maximum lateral displacement is 0.273 m, while that of the IPM and ASM is 0.665 m. The tracking precision achieved by both the IPM and the ASM falls below the desired level, primarily due to the limited control time domain. Notably, IPM performs less effectively in tracking. The RMSE of improved MPC is 0.95, ASM is 0.317, and IPM is 0.319. The IPM causes the vehicle’s front wheel angle to vary more in the 0–20 m range, which affects the stability, whereas the other two controllers show a smoother control performance.

Figure 9 reflects the computation time of the controller. It can be seen that the improved MPC algorithm takes a longer computation time in the first iteration, which is because it has a larger control time domain, which means that more dimensions of the control volume need to be solved in the iteration. Depending on the good performance of the ADMM algorithm, the average computation time of the improved MPC is still smaller than IPM and ASM. As can be seen from Table 3, the average computation time (0.0015 s) of the MPC improved by the ADMM algorithm is slightly lower than that of the ASM (0.0018 s). Compared with the IPM (0.0038 s), the average computation time of the MPC based on the ADMM algorithm is significantly reduced.

### 4.3. Comparison of Controller Computation Time as the Prediction Horizon Increases

This section of simulation discusses the advantages and disadvantages of the improved MPC based on ADMM and traditional MPC in real-time performance under different prediction time domains. When the prediction time domains are set to 8, 10, 12, 14, 16, 20 and 22, the average computation time of the three controllers is compared as shown in Table 4, and the average solution time of the control sequence by the model based on ADMM improvement is basically unchanged in different prediction time domains, as shown in Figure 10.

From Figure 10, it can be seen that as the prediction horizon increases, the computation time of the MPC based on ADMM improvement is basically stable at 0.0015 s. Compared with the active set method, the maximum average computation time is reduced by 51.6%. Compared with the interior point method, as the prediction horizon increases, the computation time of the interior point method also increases. The maximum average computation time of the MPC based on ADMM improvement is reduced by 64.3%.

The simulation results indicate that the MPC improved by the ADMM algorithm can effectively reduce the online solving time of the traditional MPC. Under the same prediction time domain and control time domain, the average computation time is reduced by 35% compared to the ASM, and reduced by 62.8% compared to the IPM. When the same control time domain is used and the prediction time domain is changed, the average computation time of the MPC based on the improved ADMM method is smaller than that of the traditional MPC. Compared with the ASM, the computation time is reduced by up to 51.6%, and compared with the IPM, the computation time is reduced by up to 64.3%. Additionally, the proposed controller has higher tracking accuracy for double lane changes with some improvement in changing lanes.

## 5. Conclusions

This paper aims to address the problem of slow online solving and the low real-time performance of traditional MPC. A combination of ADMM and MPC is used to improve the real-time performance of trajectory tracking for autonomous vehicles. The ADMM algorithm is used to solve the QP problem transformed by MPC, and the ADMM algorithm is incorporated into the receding optimization of MPC. A relaxation factor α is added to accelerate the convergence.

To validate the effectiveness of the proposed method, we established a joint simulation platform within the Carsim-Simulink environment and conducted system simulations and tests in a double-shifted lane scenario. Regarding tracking accuracy, the algorithm demonstrates significant improvements. Numerical results indicate a reduction in the tracking error by 0.217 m at the maximum transverse displacement under identical prediction and control time domains, and by 0.392 m, at the maximum transverse displacement under varying control time domains. In terms of computation time, the proposed algorithm notably enhances the real-time performance of MPC when compared to previous methods. Numerical results reveal an average computation time reduction of 64.3% compared to IPM and 51.6% compared to ASM under different control time domains. Furthermore, the ADMM algorithm exhibits stability in the computation time compared to the IPM and ASM. As the prediction time domain increases, the computation time for MPC improved by the ADMM algorithm remains nearly constant, while the computation time for the ASM fluctuates significantly and the computation time for the IPM increases.

Future research may consider integrating the proposed algorithm with an optimization toolkit to further enhance its real-time performance. Additionally, this paper does not address longitudinal control for intelligent vehicles. Therefore, the proposed algorithm could potentially be applied to the combined horizontal and longitudinal control of autonomous vehicles in the future.

## Figures and Tables

**Figure 1 sensors-23-08391-f001:**
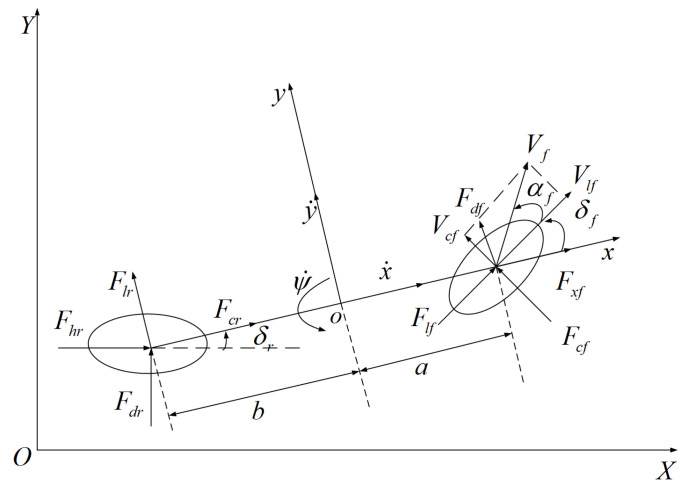
Vehicle dynamics model.

**Figure 2 sensors-23-08391-f002:**
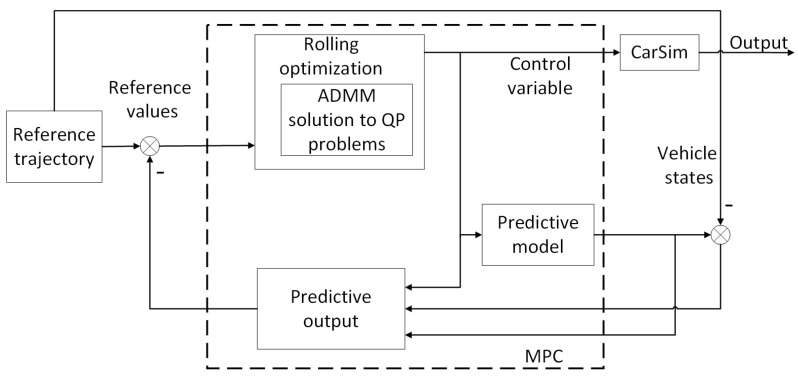
ADMM-MPC structural diagram.

**Figure 3 sensors-23-08391-f003:**
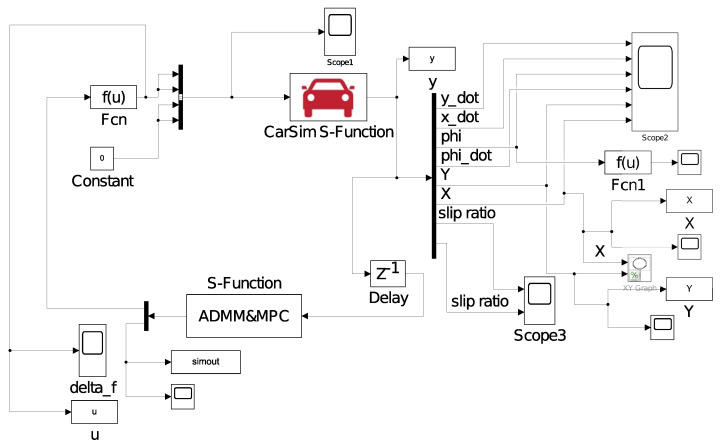
Joint simulation framework.

**Figure 4 sensors-23-08391-f004:**
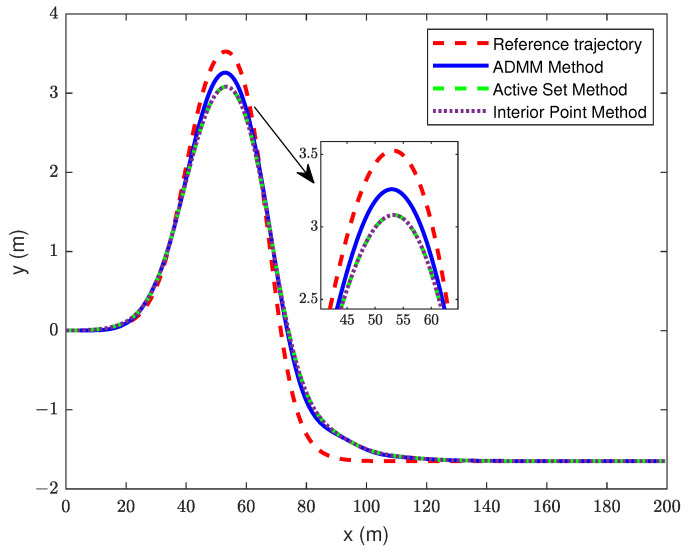
Trajectory tracking curve.

**Figure 5 sensors-23-08391-f005:**
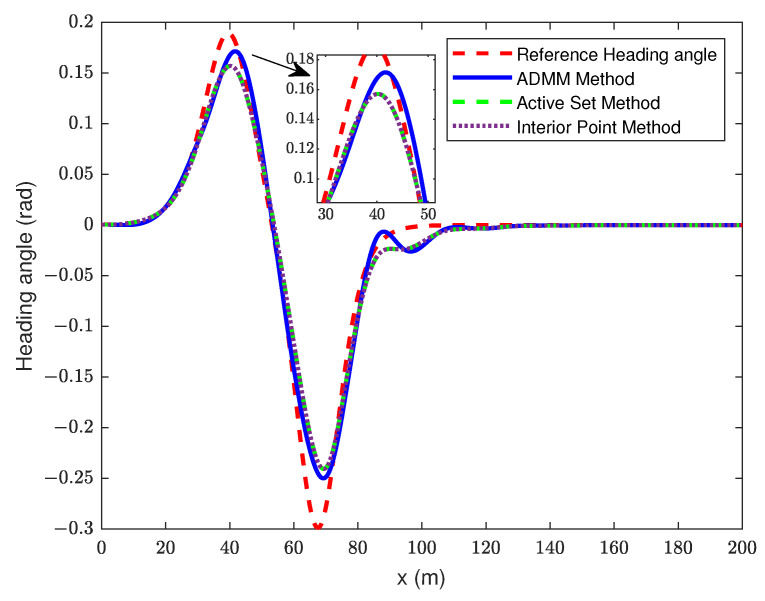
Changed curve of yaw angle.

**Figure 6 sensors-23-08391-f006:**
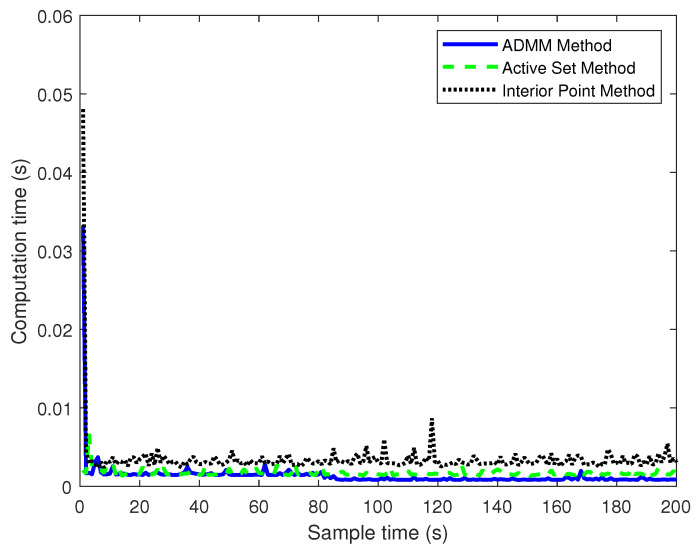
Computation time of different control methods.

**Figure 7 sensors-23-08391-f007:**
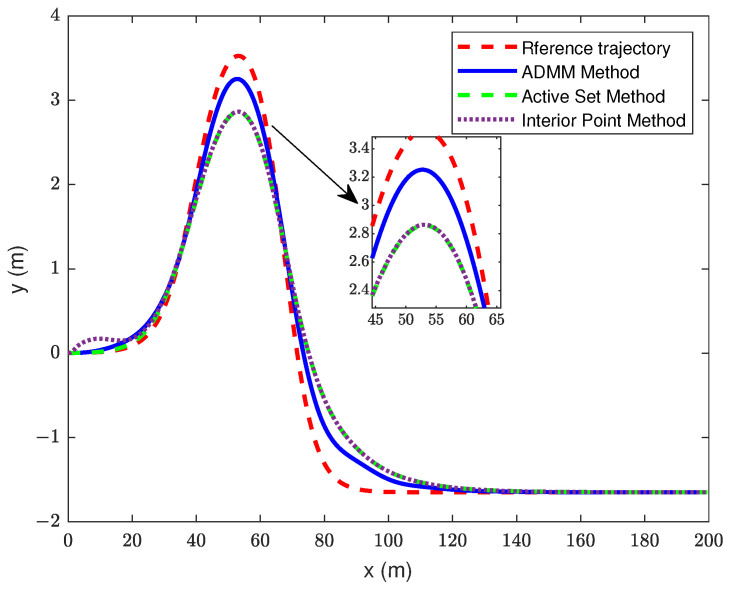
Trajectory tracking curve.

**Figure 8 sensors-23-08391-f008:**
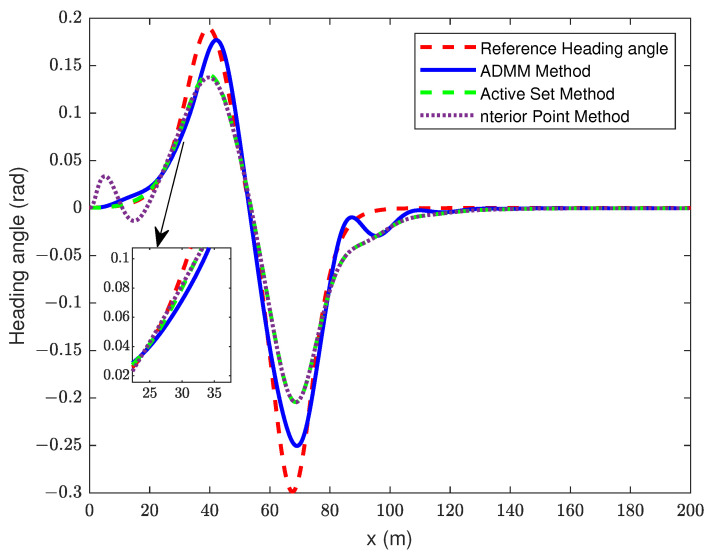
Changed curve of yaw angle.

**Figure 9 sensors-23-08391-f009:**
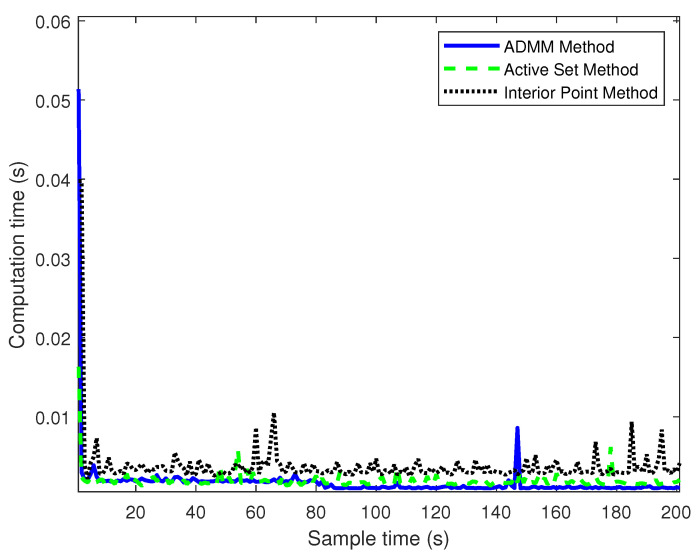
Computation time of different control methods.

**Figure 10 sensors-23-08391-f010:**
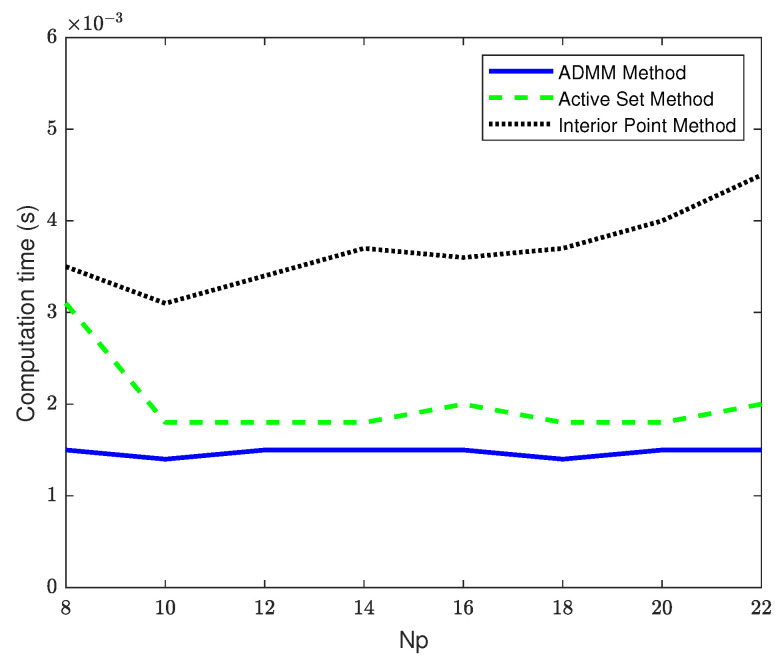
Comparison of the computation time of the three controllers in different prediction time domains.

**Table 1 sensors-23-08391-t001:** Vehicle parameters in the simulation.

Parameters	Value	Unit
vehicle weight	1723	kg
lateral moment of inertia	4331.6	kg · m${}^{2}$
roll moment of inertia	4175	kg · m${}^{2}$
distance from front axle to center of mass	1.232	m
distance from rear axle to center of mass	1.468	m
front track width	1.480	m
front and rear axle roll stiffness	2328/2653	N · m/rad
front and rear axle roll damping	47,298/37,311	N · m/rad
front wheel lateral stiffness	66,900	N/rad
rear wheel lateral stiffness	61,900	N/rad
wheel rotational inertia	0.9	kg · m${}^{2}$
rolling radius of the wheel	0.353	m

**Table 2 sensors-23-08391-t002:** Computation time of different control methods.

Method	ADMM	Active Set Method	Interior Point Method
Average computation time (s)	0.0013	0.0020	0.0035

**Table 3 sensors-23-08391-t003:** Computation time of different control methods.

Method	ADMM	Active Set Method	Interior Point Method
Average computation time (s)	0.0015	0.0018	0.0038

**Table 4 sensors-23-08391-t004:** Comparison of controller computation time in different prediction time domains.

$N_{p}$	ADMM Computation Time (s)	Active Set Method Computation Time (s)	Interior Point Method Computation Time (s)
8	0.0015	0.0031	0.0035
10	0.0014	0.0018	0.0031
12	0.0015	0.0018	0.0034
14	0.0015	0.0018	0.0037
16	0.0015	0.0020	0.0036
18	0.0014	0.0018	0.0037
20	0.0015	0.0018	0.0040
22	0.0015	0.0020	0.0045

## Data Availability

Not applicable.

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
