# Peer review of "Fast Trajectory Tracking Control Algorithm for Autonomous Vehicles Based on the Alternating Direction Multiplier Method (ADMM) to the Receding Optimization of Model Predictive Control (MPC)"

_sensors, 2023, doi:10.3390/s23208391_

Round 1

Reviewer 1 Report

>>> Summary of ideas

The paper focuses on autonomous driving and, more specifically, on unmanned vehicle trajectory tracking performance.
Researchers widely use Model Predictive Control to model the problem, so the authors used the Alternate-Direction Multiplier method to solve the Quadratic Programming problem derived from it.
Sections 2 and 3.1 introduce the background in the paper, and Section 3.2 describes the methodology (with Figure 2, on Page 9, illustrating the suggested strategy). Finally, Section 4 reports some simulations (using Simulink and Carsim) comparing the Model PRedictive Control strategy ith and without their improvement. Results are mainly compared in terms of simulation computation times.

>>> Strong points

The methodology is potentially interesting and gives some advantages regarding computation times.

>>> Weak points

The paper presents several areas for improvement that should not appear in a journal paper.

The introduction describes the state-of-the-art, but only lines 99-104 describe and introduce the new methodology. The author reports no comments on the experimental result section.

The entire Section 2, and part of Section 3 (Section 3.1), are essentially dedicated to the background. Is reporting the whole vehicle's dynamics model and the model prediction control really useful, or only some conclusion would suffice?
In this form, the background is far longer than the core section.

Results are superficially described.
The evaluation is reported only for the trajectory described in Figures 4 and 5, and those trajectories are neither defined nor motivated. Experiments are run with a single vehicle speed, and other parameters (such as the prediction frequency) are not specified (or ignored til Section 4.3).
The "tracking accuracy" (line 308 and Figures 4 and 5) is apparently evaluated only visually, and the method is not validated using standard error functions (such as RMSE, MBE, etc.).
Moreover, the performances are evaluated only in computation times; the average gain is about 25-20%, which is acceptable but not striking.

The English style should be revised, as the text contains several imprecisions or undesired aspects. Please see "Comments on the Quality of English" for a partial list of those problems.

Here are a few examples of poor English usage:
- Lines 2 and 3: "model predictive control" is repeated twice in the same phrase.
- Line 27: PID is used without definition.
- Line 63: "Taking the articulated steering tractor as the research object", uncomputed phrase.
- Line 73: "By using ... improved [26]." rephrase.
- Line 67: "space domain" is repeated.
- Line 86: "[32]; Simplifying ..." -> simplifying
- Line 113: "Vehicle dynamics model establishment" -> "Vehicle dynamics model"?
- Line 114: ", they−axis ..." -> ", the <space> y−axis ..."
- Line 246 and 253: The text goes far beyond the right margin.
- Line 290-293: The word "model" is repeated several times in the same phrase.
- Table 1: Both symbols "*" and "\cdot" are used to represent units.
- Figure 3 is practically unreadable; please increase fonts.
- Line 305: "Figure 4 ~ Figure 6" -> "Figures 4-6"
- Line 328: "Figure 7 ~ Figure 9" -> "Figures 7-9"

Author Response

1. Summary

Thank you very much for taking the time to review this manuscript. We appreciate the constructive comments. Based on each item of reviewer’s comments, the corresponding responses and actual changes implemented in detail are shown as follows.

Please find the detailed responses below and the corresponding revisions highlighted in the re-submitted files.

2. Questions for General Evaluation

Reviewer’s Evaluation

Response and Revisions

Does the introduction provide sufficient background and include all relevant references?

Yes

Are all the cited references relevant to the research?

Yes

Is the research design appropriate?

Can be improved

Are the methods adequately described?

Can be improved

Are the results clearly presented?

Must be improved

Are the conclusions supported by the results?

Must be improved

3. Point-by-point response to Comments and Suggestions for Authors

Comments 1: [The paper presents several areas for improvement that should not appear in a journal paper.]

Response 1:Thank you for your professional comment. We appreciate your suggestion. Therefore, we have re-examined the manuscript, removing literature that should not be there and adding related work on the accuracy and real-time nature of unmanned trajectory tracking. References in the original manuscript have been deleted: [30], [31]. Replaced references: [15], [17], [19], [28]. These changes can be found on page 2, lines 39, 45, 53, 70, and 78 of the revised manuscript. The corresponding revised parts are copied as follows:

(Page 2 of the revised manuscript )

.In order to improve the tracking ability of autonomous vehicles[15].

which improved the accuracy and robustness of trajectory tracking [17].

A vehicle model with steering dynamics is proposed [19].

In [28], the real-time and accuracy of vehicle trajectory tracking are improved by

[30],[31].

(Page 18 of the revised manuscript )

[15] Wang, H.; Liu, B.; Ping, X.; An, Q. Path Tracking Control for Autonomous Vehicles Based on an Improved MPC.IEEE Access 2019, 7, 161064–161073.

[17] Nan, J.; Shang, B.; Deng, W.; Ren, B.; Liu, Y. MPC-based Path Tracking Control with Forward Compensation for Autonomous Driving.IFAC-PapersOnLine 2021, 54, 443–448.

[19] Domina, Á.; Tihanyi, V. Model Predictive Controller Approach for Automated Vehicle’s Path Tracking. Sensors 2023, 23, 6862.

[19] Bai, G.; Liu, L.; Meng, Y.; Liu, S. Real-time path tracking of mobile robot based on nonlinear model predictive control. J. Agric. Mach. 2020, 51, 47–52+60.

Comments 2: [The introduction describes the state-of-the-art, but only lines 99-104 describe and introduce the new methodology. The author reports no comments on the experimental result section.]

Response 2: Thank you for your professional comment. We appreciate this comment. We have completed the changes to the introduction section to emphasize this point. In the revised manuscript, we introduced the new method in Page 2, paragraph 4, line 76 and added references [30]-[34]. Then an evaluation of experimental results section was added, which can be found on page 2, paragraph 2, lines 39, 45, 53 and page 2, paragraph 4, lines 66, 69, 72.

(Page 2 of the revised manuscript )

Alternating direction multiplier method (ADMM) is favored by many researchers because of its good scalability. [30-32].[33],[34].

This not only improves the accuracy of trajectory tracking, but also improves the steering smoothness of the vehicle.

which improved the accuracy and robustness of trajectory tracking.

However, integrating the steering system into the dynamics model can achieve optimal performance, but leads to higher computational requirements.

The above methods improve the tracking accuracy to some extent, but do not consider the complexity of MPC calculation.

However, it considers the longitudinal motion of the vehicle as well as the energy saving.

Reducing the control frequency domain can meet the real-time requirements, but the error is slightly greater than that of reducing the control step.

(Page 18-19 of the revised manuscript )

[30] Cheng, Z.; Ma, J.; Zhang, X.; Lee, T. Semi-Proximal ADMM for Model Predictive Control Problem with Application to a UAV System. In proceedings of the 2020 20th International Conference on Control, Automation and Systems (ICCAS), Busan, Korea (South), 13-16 October 2020; pp. 82–87.

[31] Li, X.; Tyagi, A. Block-Active ADMM to Minimize NMF with Bregman Divergences. Sensors 2023,23, 7229.

[32]Toyoda, M.; Tanaka, M. An analysis of hot-started ADMM for linear MPC. IET Control. Theory Appl 2021, 15, 1999–2016.

[33] Hong, M.; Lou, Z.; Razaviyayn, M. Convergence analysis of alternating direction method of multipliers for a family of nonconvex problems. In proceedings of the 2015 IEEE International Conference on Acoustics, Speech and Signal Processing (ICASSP), South Brisbane, QLD, Australia, 19-24 April 2015; pp. 3836–3840.

[34]Ma, J.; Cheng, X.; Zhang, X.; Tomizuka, M.; Lee, T. Alternating Direction Method of Multipliers for Constrained Iterative LQR in Autonomous Driving. IEEE trans Intell Transp Syst 2022, 23,  23031–23042.

Comments 3: [The entire Section 2, and part of Section 3 (Section 3.1), are essentially dedicated to the background. Is reporting the whole vehicle's dynamics model and the model prediction control really useful, or only some conclusion would suffice?
In this form, the background is far longer than the core section
.]

Response 3: Thank you for your professional comment. We appreciate your suggestion. We have re-examined the revised manuscript and have rationalized and streamlined the vehicle dynamics modeling and model predictive control sections, deleting some unnecessary derivation processes and retaining important conclusions. For example, we removed the unnecessary derivations of Eq. (2), Eq. (4), Eq. (9), Eq. (10), Eq. (11), and Eq. (13) from the original manuscript.

Comments 4: [Results are superficially described.
The evaluation is reported only for the trajectory described in Figures 4 and 5, and those trajectories are neither defined nor motivated. Experiments are run with a single vehicle speed, and other parameters (such as the prediction frequency) are not specified (or ignored til Section 4.3).
The "tracking accuracy" (line 308 and Figures 4 and 5) is apparently evaluated only visually, and the method is not validated using standard error functions (such as RMSE, MBE, etc.).
Moreover, the performances are evaluated only in computation times; the average gain is about 25-20%, which is acceptable but not striking.
]

Response 4: Thank you for your professional suggestion. We appreciate your suggestion. We have given the definition of double shift line and added the mathematical expression for double shift line on page 11, paragraph 1, line 289 in the revised manuscript. The selection of the parameter prediction time domain and control time domain is explained on page 11, paragraph 2, line 297. Specific values are given in Sections 4.1 and 4.2.

The RMSE of the trajectory tracking accuracy is given on page 12, paragraph 2, and the tracking error values at the maximum lateral displacement are given for the three methods.

The real-time performance of the proposed method has been shown in Tables 2, 3, and 4, but the results may be not striking, thanks to your comments, which make we realize that there is still room for improvement in the proposed method and point out the direction of the next improvement.

(Page 11 of the revised manuscript )

In the ordinary vehicle driving test, the double-shift condition is the test section with  high frequency. Many scholars have also used it to test the track-tracking capabilities of autonomous vehicles. The desired trajectory is given by:

The initial parameters of the algorithm are set as road adhesion coefficient µ=0.85, v=20 m/s, relaxation factor α=1.7, and the penalty factor ρ is a decreasing sequence with respect to time. The calculation time of the three algorithms is compared in the same and different prediction time domain and control time domain. The calculation time of the three algorithms is compared in the same and different prediction time domain and control time domain.

(Page 12 of the revised manuscript )

4.1. Comparison of controllers under the same prediction and control horizon(Np = 11,Nc = 6)

In this section of simulation, the control time domain of the MPC based on ADMM improvement is set to 10, and the control time domain of the traditional MPC is 4.

We can also get the improved MPC shows better tracking accuracy, the proposed controller has a tracking error of 0.266m at the maximum lateral displacement, while the other two controllers have a tracking error of 0.443m. During the whole control time, we use the trajectory data driven by the vehicle and the expected trajectory data to obtain the root mean square error(RMSE) of each controller. The RMSE for improved MPC is 0.193, both other controllers is 0.231. The RMSE formula is shown below:

It is worth noting that the driving comfort may be reduced due to the change in the heading angle, but it can quickly track the reference yaw angle afterwards.

(Page 13 of the revised manuscript )

The error of the improved MPC at the maximum lateral displacement is 0.273m, while that of the IPM and ASM is 0.665m. The tracking accuracies of IPM and ASM are not satisfactory due to the short control time domain, with IPM exhibiting worse tracking. The RMSE for improved MPC is 0.195, for ASM is 0.317, and for IPM is 0.319. The IPM causes the vehicle’s front wheel angle to vary more in the 0-20m range, which affects stability, whereas the other two controllers show smoother control performance.

4. Response to Comments on the Quality of English Language

Point 1:Lines 2 and 3: "model predictive control" is repeated twice in the same phrase.

Response 1: Thank you for your suggestion. We appreciate your suggestion. We have modified it, as can be seen on page 1, line 1.

The corresponding revised parts are copied as follows:

In order to improve the real-time performance of trajectory tracking of unmanned vehicles, this paper applies the alternating direction multiplier method (ADMM) to the rolling optimization of model predictive control (MPC), which improved the computational speed of the algorithm.

Point 2:Line 27: PID is used without definition.

Response 2: Thank you for your suggestion. We appreciate your suggestion. We have revised that sentence, as can be seen on page 1, Paragraph 2, line 26. It is copied as follows:

Proportional integral derivative (PID)

Point 3:Line 63: "Taking the articulated steering tractor as the research object", uncomputed phrase.

 Response 3: Thank you for your suggestion. We appreciate your suggestion. We have revised this reference.

Point 4:Line 73: "By using ... improved [26]." rephrase.

Response 4: Thank you for your suggestion. We appreciate this suggestion. We've rewritten the sentence, as can be seen on page 2, Paragraph 4, line 64. It is copied as follows:

A genetic algorithm is used to compute the optimal time domain parameters for real-time vehicle speed and road conditions, which improves the real-time performance of the controller[26].

Point 5:Line 67: "space domain" is repeated.

Response 5: Thank you for your suggestion. We appreciate your suggestion. We deleted the redundant phrase "space domain", as can be seen on page 2, Paragraph 4, line 68. It is copied as follows:

a nonlinear space-domain MPC (SMPC) is proposed to accelerate the nonlinear SMPC computation by generating thermal initialization and subsequently forming SMPC-RTI.

Point 6:Line 86: "[32]; Simplifying ..." -> simplifying

Response 6: Thank you for your suggestion. We appreciate your suggestion. We've replaced "Simplifying" with "simplifying",as can be seen on page 3, Paragraph 5, line 91. It is copied as follows:

simplifying the vehicle model can also improve the real-time performance of the controller

Point 7:Line 113: "Vehicle dynamics model establishment" -> "Vehicle dynamics model"?

Response 7: Thank you for your suggestion. We appreciate your suggestion. We've made changes to it, as can be seen on page 3, line 116. It is copied as follows:

2.1. Vehicle dynamics model

Point 8:Line 114: ", they−axis ..." -> ", the <space> y−axis ..."

Response 8: Thank you for your suggestion. We appreciate your suggestion. We've made changes to it, as can be seen on page 3, line 122. It is copied as follows:

The x − axis represents the longitudinal axis in the vehicle coordinate system, the y − axis represents the lateral axis in the vehicle coordinate system

Point 9: Line 246 and 253: The text goes far beyond the right margin.

Response 9: Thank you for your suggestion. We appreciate your suggestion. We have modified the formula that is out of bounds, as can be seen on page 9, line 240 and line 246.

Point 10: Line 290-293: The word "model" is repeated several times in the same phrase.

Response 10: Thank you for your suggestion. We appreciate your suggestion. We didn't find the word "model" in the appropriate place, so we guessed that it should be "method", so we abbreviated the mentioned method, as can be seen on page 10, line 283 to 288. It is copied as follows:

To verify the effectiveness of the proposed algorithm, a joint simulation platform of Simulink and CarSim was built to simulate and validate the designed controller. The ASM and the IPM are commonly used methods for solving traditional MPC problems, and this paper compares and analyzes the proposed algorithm with ASM and IPM. The processor parameters of the laptop used in the simulation are AMD Ryzen 7 5700U with Radeon Graphics 1.80 GHz. The vehicle parameters used in simulation are shown in Table 1.

Point 11: Table 1: Both symbols "*" and "\cdot" are used to represent units.

Response 11: Thank you for your suggestion. We appreciate your suggestion. We scrutinized Table 1 and replaced all "*" with "\cdot", as can be seen Table 1(page 11).

Point 12: Figure 3 is practically unreadable; please increase fonts.

Response 12: Thank you for your suggestion.We've increased the font size of Figure 3 to make sure it can be seen. Please see Figure 3 on page 11 for details.

Point 13:Line 305: "Figure 4 ~ Figure 6" -> "Figures 4-6"

Response 13: Thank you for your suggestion. We appreciate this suggestion. We have replaced "Figure 4 ~ Figure 6" with "Figures 4-6". It can be viewed on page 12, paragraph 1, line 304. It is copied as follows:

The simulation results are shown in Figures 4-6.

Point 14:Line 328: "Figure 7 ~ Figure 9" -> "Figures 7-9"

Response 14: Thank you for your suggestion. We appreciate your suggestion. We have replaced "Figure 7 ~ Figure 9" with "Figures 7-9". It can be viewed on page 14, paragraph 2, line 338. It is copied as follows:

The simulation results are shown in Figures 7-9.

5. Additional clarifications

We modified Figures 4-6, Figures 7-9, Tables 2, 3, and 4 in the revised draft by replacing "Effective Set Method" with "Active Set Method" in the label of each figure. The same change was made in the Table 2, 3, 4.

Thanks again.

Reviewer 2 Report

Dear authors, thank you for your work.
Overall, the article shows good results, but should be thoroughly revised. Here are some comments:

- The vehicle model should be better introduced. It is not clear what kind of vehicle is considered. Fig. 1 indicates that the rear wheel is rotatable? So no classic passenger car is considered? Also the labels are hard to see and the angle orientation are not visible.

-References are missing for some statements. Thus, a number of (alleged) facts are presented without supporting their origin with a reference.  

- Check the references, there seems to be occasional references to the wrong equation.

- The costs (20) only weight the trajectory and the input. What about the other states? Surely they don't matter?

- Line 221 "are the linear inequality constraints"?

- Fig. 2: Feedback correction? This is not explained and has no output.

- Table 1: star vs. dot

- You describe as a comparison methods, but not the solvers. This has a significant influence on the results of all methods.

Dear authors, thank you for your work.
The article should be thoroughly revised in terms of language. There are a lot of errors (capitalization, doctrinal marks, repetitions, use of wrong terms). Also, for a good style, the equations should be incorporated into the sentence structure.

Author Response

1. Summary

Thank you very much for taking the time to review this manuscript. We appreciate the constructive comments. Based on each item of reviewer’s comments, the corresponding responses and actual changes implemented in detail are shown as follows.

Please find the detailed responses below and the corresponding revisions highlighted in the re-submitted files.

2. Questions for General Evaluation

Reviewer’s Evaluation

Response and Revisions

Does the introduction provide sufficient background and include all relevant references?

Can be improved

Are all the cited references relevant to the research?

Must be improved

Is the research design appropriate?

Can be improved

Are the methods adequately described?

Can be improved

Are the results clearly presented?

Yes

Are the conclusions supported by the results?

Can be improved

3. Point-by-point response to Comments and Suggestions for Authors

Comments 1: [The vehicle model should be better introduced. It is not clear what kind of vehicle is considered. Fig. 1 indicates that the rear wheel is rotatable? So no classic passenger car is considered? Also the labels are hard to see and the angle orientation are not visible.]

Response 1: Thank you for pointing this out. We appreciate your suggestion. Therefore, We present a dynamic model of the vehicle in the revised manuscript as a model of a bicycle with a steerable front wheel. This can be found on page 3, lines 117 through 120. We enhanced the font in Figure 1 and labeled the angular direction of wheel rotation. We have revised the text to address your concerns and hope that it is now clearer. The corresponding revised parts are copied as follows:

(Page 3 of the revised manuscript )

 In this model, we do not consider the effects of suspension and aerodynamics, the state quantity is defined as and the control input vector is the front wheel angle, defined as .

Comments 2: [References are missing for some statements. Thus, a number of (alleged) facts are presented without supporting their origin with a reference. ]

Response 2: Thank you for your professional comment. We appreciate this comment. Therefore, We have thoroughly examined the revised draft, and have filled in references that may be missing. For example, on page 3, line 91, "simplifying the vehicle model can also improve the real-time performance of the controller, but it may not achieve ideal control effects in complex road conditions " reference [28] was added, and on page 3, line 117, reference [38] was added for vehicle dynamics modeling. The corresponding revised parts are copied as follows:

(Page 3 of the revised manuscript )

 simplifying the vehicle model can also improve the real-time performance of the controller, but it may not achieve ideal control effects in complex road conditions[28].

To ensure that the vehicle follows its desired trajectory, the bicycle model in [38] is

borrowed to represent the vehicle dynamics.

(Page 18 of the revised manuscript )

[28] Bai, G.; Liu, L.; Meng, Y.; Liu, S. Real-time path tracking of mobile robot based on nonlinear model predictive control. J. Agric. Mach. 2020, 51, 47–52+60.

(Page 18 of the revised manuscript )

[38] Lin, X.; Tang, Y.; Zhou, B. Improved Model Predictive Control Path Tracking Strategy Based an Online Updating Algorithm With Cosine Similarity and a Horizon Factor.IEEE trans Intell Transp Syst 2022, 23, 12429–12438.

Comments 3: [Check the references, there seems to be occasional references to the wrong equation.]

Response 3:Thank you for your professional comment. We appreciate your suggestion. Therefore, We have checked all references and found that the format of references [6], [7], [9], [10], [12], [13], [14], [15], [17], [21], [22], [23], [24], [25], [28], [29], [32], [33], [34] are incorrect. They have all been revised in the revised manuscript. The corresponding revised parts are copied as follows:

(Page 17 of the revised manuscript )

 [6] Li, S.; Wang, G.; Zhang, B.; Yu Z.; Cui, G. Vehicle Yaw Stability Control at the Handling Limits Based on Model Predictive Control. Int. J. Automot. Technol. 2020, 21, 361–370.

(Page 18 of the revised manuscript )

[7] Qi, L.; Jiao, X.; Wang, Z. Trajectory Tracking Control of Intelligent Vehicle Based on DDPG Method of Reinforcement Learning. China J. Highw. Transp. 2021, 34, 335–348.

[9] Kanchwala, H.; Bezerra, V.I.; Aouf, N. Cooperative path-planning and tracking controller evaluation using vehicle models of varying complexities. Proc. Inst. Mech. Eng. Part C J. Mech. Eng. Sci. 2020, 235, 2877–2896.

[10] Choi, W.; Nam, H.; Kim, B.; Ahn, C. Model Predictive Control for Evasive Steering of Autonomous Vehicle. Int. J. Automot. Technol. 2019, 20, 1033–1042.

[12] Jiang, C.; Zhai, J.; Tian, H.; Wei, C.; Hu, J. Approximated Long Horizon MPC with Hindsight for Autonomous Vehicles Path Tracking. In Proceedings of the 2020 3rd International Conference on Unmanned Systems (ICUS), Harbin, China, 27-28, November 2020; pp. 696–701.

[13] Mata, S.; Zubizarreta, A.; Pinto, C.Robust tube-based model predictive control for lateral path tracking.IEEE Trans. Intell. Veh. 2019, 4, 569–577.

[14] Tang, Z.; Xu, X.; Wang, F. Coordinated control for path following of two-wheel independently actuated autonomous ground vehicle.IET Intell. Transp. Syst. 2019, 13, 628–635.

[15] Wang, H.; Liu, B.; Ping, X.; An, Q. Path Tracking Control for Autonomous Vehicles Based on an Improved MPC.IEEE Access 2019, 7, 161064–161073.

[17] Nan, J.; Shang, B.; Deng, W.; Ren, B.; Liu, Y. MPC-based Path Tracking Control with Forward  Compensation for Autonomous Driving.IFAC-PapersOnLine 2021, 54, 443–448.

[21] Xu, F.; Zhang, J.; Hu, Y.; Qu, T.; Qu, Y.; Liu, Q. Lateral and longitudinal coupling real-time predictive controller for intelligent vehicle path tracking. J. Jilin Univ. (Eng. Technol. Ed.) 2021, 51, 2287–2294.

[22] Shekhar, R.C.; Manzie, C. Optimal move blocking strategies for model predictive control. Automatica 2015, 61, 27–34.

[23] Leng, Y.; Zhao, S. Explicit Model Predictive Control for Intelligent Vehicle Lateral Trajectory Tracking. J. Syst. Simul. 2021, 33, 1177–1187.

[24] Kayacan, E.; Kayacan, E.; Ramon, H.; Saeys, W. Learning in Centralized Nonlinear Model Predictive Control: Application to an Autonomous Tractor-Trailer System. IEEE Trans. Control Syst. Technol. 2014, 23, 197–205.

[25] Pas, P.; Schuurmans M.; Patrinos, P. Alpaqa: A matrix-free solver for nonlinear MPC and large-scale nonconvex optimization. In Proceedings of the 2022 European Control Conference (ECC), London, UK, 12-15, July 2022; pp. 417–422.

[28] Bai, G.; Liu, L.; Meng, Y.; Liu, S. Real-time path tracking of mobile robot based on nonlinear model predictive control. J. Agric. Mach. 2020, 51, 47–52+60.

[29] Alcalá, E.; Puig, V.; Quevedo, J. LPV-MPC Control for Autonomous Vehicles. IFAC-PapersOnLine 2019, 52, 106–113.

[32] Toyoda, M.; Tanaka, M. An analysis of hot-started ADMM for linear MPC. IET Control. Theory Appl 2021, 15, 1999–2016.

[33] Hong, M.; Lou, Z.; Razaviyayn, M. Convergence analysis of alternating direction method of multipliers for a family of nonconvex problems. In proceedings of the 2015 IEEE International Conference on Acoustics, Speech and Signal Processing (ICASSP), South Brisbane, QLD, Australia, 19-24 April 2015; pp. 3836–3840.

[34] Ma, J.; Cheng, X.; Zhang, X.; Tomizuka, M.; Lee, T. Alternating Direction Method of Multipliers for Constrained Iterative LQR in Autonomous Driving. IEEE trans Intell Transp Syst 2022, 23, 23031–23042.

Comments 4: [The costs (20) only weight the trajectory and the input. What about the other states? Surely they don't matter?]

Response 4:Thank you for your professional comment. We appreciate your suggestion. We have revised the sentence on page 6, paragraph 2, line 177. In the objective function, the output is not only the vehicle's position information, but also the vehicle's traverse angle, speed and other information. The corresponding revised parts are copied as follows:

(Page 6 of the revised manuscript )

   represents the actual output of the system, represents the reference output

Comments 5: [Line 221 "are the linear inequality constraints"?]

Response 5:Thank you for your professional comment. We appreciate your suggestion. The constraints of Eq. (17) are the equational constraints. We 've corrected the typo. The corresponding revised parts are copied as follows:

(Page 7 of the revised manuscript )

   are the linear equality constraints that the problem needs to satisfy.

Comments 5:[Fig. 2: Feedback correction? This is not explained and has no output.]

Response 5:Thank you for your professional comment. We appreciate your suggestion. We have modified the three main modules of the MPC in Figure 2. Please see Figure 2 on page 8 for details.

Comments 6:[Table 1: star vs. dot.]

Response 6:Thank you for your suggestion. We appreciate your suggestion. We scrutinized Table 1 and replaced all "*" with "\cdot", as can be seen Table 1(page 11).

Comments 7:[You describe as a comparison methods, but not the solvers. This has a significant influence on the results of all methods.]

Response 7:Thank you for your suggestion. We appreciate your suggestion. The methods we are comparing are with existing methods. The real-time performance of the proposed method has been shown in Tables 2, 3, and 4. We did not take into account the comparison with the solver, therefore, we thank you for your suggestion to realize that there is room for improvement in the proposed method.

4. Response to Comments on the Quality of English Language

Point 1:The article should be thoroughly revised in terms of language. There are a lot of errors (capitalization, doctrinal marks, repetitions, use of wrong terms). Also, for a good style, the equations should be incorporated into the sentence structure.

Response 1: Thank you for your suggestion. We appreciate this comment. We rechecked the revised manuscript and found some errors, for example, in line 27 of the original manuscript, the use of PID is not defined; line 73: "By using ... improved [26]." needs to be rewritten; line 67: "space domain" is repeated. line 86 "Simplifying" should be changed to "simplifying", line 90 "Quadratic Programming" should be initialized in lower case, missing space in line 114, etc. We have changed them in the revised draft. For the formula section, we have integrated each formula into the sentence.

5. Additional clarifications

We modified Figures 4-6, Figures 7-9, Tables 2, 3, and 4 in the revised draft by replacing "Effective Set Method" with "Active Set Method" in the label of each figure. The same change was made in the Table 2, 3, 4.

Thanks again.

Round 2

Reviewer 1 Report

The authors fixed many problems of the previous version.
Although they only partially replied to some of my previous observations (for example, I consider the experimental result section still weak), the new version may be acceptable for publication.

I suggest a moderate editing of the English language to improve the readability and quality of the paper.

Author Response

1. Summary

Thank you very much for taking the time to review this manuscript. We appreciate the constructive comments. Based on each item of reviewer’s comments, the corresponding responses and actual changes implemented in detail are shown as follows.

Please find the detailed responses below and the corresponding revisions highlighted in the re-submitted files.

2. Questions for General Evaluation

Reviewer’s Evaluation

Response and Revisions

Does the introduction provide sufficient background and include all relevant references?

Yes

Are all the cited references relevant to the research?

Can be improved

Is the research design appropriate?

Can be improved

Are the methods adequately described?

Can be improved

Are the results clearly presented?

Must be improved

Are the conclusions supported by the results?

Must be improved

3. Point-by-point response to Comments and Suggestions for Authors

Comments 1: [The authors fixed many problems of the previous version.
Although they only partially replied to some of my previous observations (for example, I consider the experimental result section still weak), the new version may be acceptable for publication.]

Response 1:Thank you for your professional comment. We appreciate your suggestion. Thank you for your previous professional feedback and affirmation of our work. As a result, we have expanded the description of the experimental results section and have also outlined potential improvements that can be made to the method in the future. These changes can be found on page 17, from line 377 to line 400 of the revised manuscript. The corresponding revised parts are copied as follows:

(Page 17 of the revised manuscript )

This paper aims to address the problem of slow online solving and low real-time performance of traditional MPC. A combination of ADMM and MPC is used to improve the real-time performance of trajectory tracking for autonomous vehicles. The ADMM algorithm is used to solve the QP problem transformed by MPC, and the ADMM algorithm is incorporated into the receding optimization of MPC. A relaxation factor α is added to speed up the convergence.

To validate the effectiveness of the proposed method, we established a joint simulation  platform within the Carsim-Simulink environment and conducted system simulations and tests in a double-shifted lane scenario. Regarding tracking accuracy, the algorithm demonstrates significant improvements. Numerical results indicate a reduction of the tracking error by 0.217m at the maximum transverse displacement under identical prediction and control time domains, and by 0.392m at the maximum transverse displacement under varying control time domains. In terms of computation time, the proposed algorithm notably enhances the real-time performance of MPC when compared to previous methods. Numerical results reveal an average computation time reduction of 64.3% compared to IPM and 51.6% compared to ASM under different control time domains. Furthermore, the ADMM algorithm exhibits stability in computation time compared to the IPM and ASM. As the prediction time domain increases, the computation time for MPC improved by the ADMM algorithm remains nearly constant, while the computation time for the ASM fluctuates significantly and the computation time for the IPM increases.

Future research may consider integrating the proposed algorithm with an optimization  toolkit to further enhance its real-time performance. Additionally, this paper does not address longitudinal control for intelligent vehicles. Therefore, the proposed algorithm could potentially be applied to the combined horizontal and longitudinal control of autonomous vehicles in the future.

4. Response to Comments on the Quality of English Language

Point 1: I suggest a moderate editing of the English language to improve the readability and quality of the paper.

Response 1: Thank you for your suggestion. We appreciate your suggestion.

We conducted a comprehensive review of the English grammar and vocabulary in the manuscript, making modifications where errors were identified. Additionally, we improved the readability of longer sentences. We also addressed any potential issues in the figures and tables throughout the paper.

Thanks again.

Reviewer 2 Report

All my comments have been adequately addressed.

There are only minor spelling errors.

Author Response

1. Summary

Thank you very much for taking the time to review this manuscript. We appreciate the constructive comments. Based on each item of reviewer’s comments, the corresponding responses and actual changes implemented in detail are shown as follows.

Please find the detailed responses below and the corresponding revisions highlighted in the re-submitted files.

2. Questions for General Evaluation

Reviewer’s Evaluation

Response and Revisions

Does the introduction provide sufficient background and include all relevant references?

Can be improved

Are all the cited references relevant to the research?

Can be improved

Is the research design appropriate?

Can be improved

Are the methods adequately described?

Can be improved

Are the results clearly presented?

Yes

Are the conclusions supported by the results?

Can be improved

3. Point-by-point response to Comments and Suggestions for Authors

Comments 1: [All my comments have been adequately addressed.]

Response 1: Thank you for your recognition of our previous work. In the last round of revisions, We learned a lot from your professional feedback. Once again, thank you for your generous and valuable professional advice.

4. Response to Comments on the Quality of English Language

Point 1:There are only minor spelling errors.

Response 1: Thank you for your professional comment. We appreciate this comment. Therefore, We conducted a comprehensive review of the manuscript and made revisions to areas where errors could potentially occur. For instance, we replaced unclear sentences, such as the one on page 1, line 12, and made similar modifications on page 2, line 45, and so on. For other changes, please refer to the highlighted sections in the revised manuscript.

5. Additional clarifications

We have optimized Figure 2 to make it appear more aesthetically pleasing.

Thanks again.
